# Obtaining and Properties of a Photocatalytic Composite Material of the “SiO_2_–TiO_2_” System Based on Various Types of Silica Raw Materials

**DOI:** 10.3390/nano11040866

**Published:** 2021-03-29

**Authors:** Valeria Strokova, Ekaterina Gubareva, Yulia Ogurtsova, Roman Fediuk, Piqi Zhao, Nikolai Vatin, Yuriy Vasilev

**Affiliations:** 1V.G. Shukhov Belgorod State Technological University, 308012 Belgorod, Russia; vvstrokova@gmail.com (V.S.); ekaterina.bondareva@rambler.ru (E.G.); ogurtsova.y@yandex.ru (Y.O.); 2Polytechnic Institute, Far Eastern Federal University, 690922 Vladivostok, Russia; 3University of Jinan, Jinan 250000, China; mse_zhaopq@ujn.edu.cn; 4Peter the Great St. Petersburg Polytechnic University, 195251 St. Petersburg, Russia; vatin@mail.ru; 5Moscow Automobile and Road Construction University, 125319 Moscow, Russia; yu.vasilev@madi.ru

**Keywords:** sol–gel synthesis, titanium dioxide, silica support, photocatalysis, composite material, degradation of rhodamine B

## Abstract

Compositions and technology for obtaining a photocatalytic composite material (PCM) by deposition of titanium dioxide particles synthesized by the sol–gel method on a silica support of various types (microsilica, gaize and diatomite) have been developed. The properties (chemical and mineral composition, dispersion, specific surface area, porosity, ζ-potential, acid–base properties, and microstructure) of microsilica, gaize and diatomite were studied to assess the effectiveness of using a photocatalytic agent as a carrier. In terms of specific viscosity (η_sp_ = 45), the concentration of the precursor (tetrabutoxytitanium—TBT) is set at 22 vol. % in a solvent (ethanol), at which it is possible to obtain the maximum amount of dissolved film oligomer without the formation of an aggregate-like precipitate. Modification of the reaction mixture (precursor: ethanol = 1:3) by replacing part of the solvent with a Span-60 surfactant/TBT = 1–1.1 made it possible to obtain polydisperse titanium dioxide particles with peak sizes of 43 nm and 690 nm according to laser granulometry data. Taking into account the interaction of titanium complexes with the surface of a silica support, a phenomenological model of the processes of structure formation of a photocatalytic composite material is proposed. By the value of the decomposition of rhodamine B, the photocatalytic activity of the developed composite materials was determined: PCM based on diatomite—86%; PCM based on microsilica—85%; PCM based on gaize—57%.

## 1. Introduction

Among the numerous semiconductor photocatalysts, titanium dioxide (mainly in the polymorphic modification of anatase) occupies a special place due to its high photocatalytic activity, physical and chemical stability, non-corrosiveness, non-toxicity, availability and relatively low price [1,2,3,4]. One of the main goals of scientific research aimed at increasing the efficiency of the practical use of anatase is to improve its interaction with electromagnetic radiation (ultraviolet and visible) and adsorbed pollutants as a factor of high photocatalytic activity. This can be realized, for example, by: modifying the anatase structure by doping, achieving the nanoscale of its crystals, creating photocatalytic composite materials, etc. [5,6,7]. At the same time, rational methods of technological solutions for these problems are selected by taking into account the field of application of the photocatalyst, including the existing technology for obtaining and using the final material with its use, in order to ensure efficiency, economy and high self-cleaning ability.

In this regard, from the point of view of the effectiveness of using anatase as a photocatalyst for construction, taking into account the large-scale industry of building materials, it is advisable to study the possibility of creating photocatalytic composite materials (PCMs). This is due to the need to: reduce the cost of implementing the self-cleaning process in building composites, simplify and increase the uniformity of the distribution of the nanoscale photocatalyst in the near-surface layer and increase the surface area of the photocatalyst interacting with electromagnetic radiation and pollutant molecules. The research results show that an effective PCM for construction can be obtained by depositing titanium dioxide on a silica support [8,9,10,11,12,13].

A review of methods for preparing composite materials, in which SiO_2_ is used as a carrier in one state or another, makes it possible to note that the sol–gel technology is the most effective and well controlled [14,15,16]. The morphology of the synthesized particles, their size and content and polymorphic modification of the final product can be controlled and regulated at all stages of the synthesis. Additionally, the advantages of the sol–gel technology are low temperatures of processes and homogeneity at the molecular level, which make it possible to obtain materials with desired properties, which is important for the PCM of the SiO_2_–TiO_2_ system [17,18,19,20].

When obtaining titanium dioxide particles in the process of sol–gel synthesis, the features of the titanium-containing precursor and the composition of the reaction mixture are taken into account, the choice of which determines the conditions for the synthesis and the need to introduce certain reagents into the reaction mixture. The components used, as well as the synthesis conditions, can affect the morphology of titanium dioxide particles and their interaction with the silica support, which determines the photocatalytic activity of the composite material.

Based on the results of studies of composite materials by scientists from different countries, it was found that the combined use of TiO_2_ and SiO_2_ has a positive effect on the properties of titanium dioxide, increasing the temperature of the phase transition of anatase to rutile [21]. The resulting materials are characterized by increased photocatalytic activity, assessed by the degree of the degradation of pollutants and an increased degree of surface hydrophilicity, which facilitates the removal of decomposition products from the composite surface with water. Thus, the improvement of the characteristics of a photocatalyst deposited on a silica support determines a wide range of practical applications of the photocatalytic composite material.

The aim of this work was to develop the composition of the raw mixture and the technology for the synthesis of a photocatalytic composite material with a self-cleaning function, which ensure the formation of nanosized anatase particles uniformly distributed on the surface of the silica support and a high photocatalytic activity of the final material.

## 2. Materials and Methods

### 2.1. Sol–Gel Synthesis Raw Materials

To obtain nanosized particles of dioxide, the organo-inorganic titanium-containing precursor tetrabutoxitanium (TBT) C_16_H_36_O_4_Ti (Promkhimperm, Permian, Russia), containing 14.3 wt. % titanium, was used.

The processes of hydrolysis and polycondensation were carried out in an alcoholic environment, which included: butyl alcohol C_4_H_10_OH (Component-Reagent, Moscow, Russia), isopropyl alcohol C_3_H_8_O (EKOS-1, Moscow, Russia) and ethyl alcohol C_2_H_5_OH (RFK, Moscow, Russia), using nonionic surface-active substances (surfactants): Span-60 (hydrophilic–lipophilic balance (HLB = 4.7), Span-80 (HLB = 4.3) (Sigma-Aldrich Chemie GmbH, St. Louis, MO, USA) and Span-83 (HLB = 3.7) (Alfa Aesar Thermo Fisher (Kandel) GmbH, Ward Hill, MA, USA), as well as Tween-80 (HLB = 15) (Oleon NV, Oelegem, Belgium). These surfactants were introduced into the reaction medium both individually and in the form of mixtures. To prepare a mixture of surfactants with a given HLB value, the quantitative values of components *A* and *B* with known HLB were calculated using Formula (1):(1)$$(A)\%=\frac{(X-HLB_{B})\times 100}{HLB_{A}-HLB_{B}};(B)\%=100-(A)\%$$
where (*A*)%, (*B*)% is the amount of component *A* and *B*, respectively; *X* is the set HLB value for the mixture; *HLB_A_* and *GLB_B_* are the HLB values of component *A* and component *B*, respectively.

As a carrier of anatase in the composition of a photocatalytic composite material, the possibility of using ready-made commercial silica raw materials of various genesis was investigated: natural biogenic and chemo-biogenic diatomite of the Inzenskoye deposit (Ulyanovsk, Russia), gaize of the Alekseevsky deposit (Mordovia, Russia) and technogenic microsilica (Lipetsk, Russia).

To prepare samples for studying the photocatalytic activity of the synthesized composite materials, white cement CEM I 52.5 R (Adana Cement, Adana, Turkey) was used. An industrial photocatalyst AEROXIDE TiO_2_ P25 (Evonik Industries AG, Essen, Germany) with D_av_ = 21 nm, S_sp_ = 50 m^2^/g, TiO_2_ more than 95 wt. % was used as a control sample, with % represented by 92% anatase and 7% rutile.

As a result of the sol–gel synthesis of nanosized titanium dioxide particles on the surface of silica supports, three types of powdery photocatalytic composite materials (PCMs) were obtained: PCM_diatomite_, PCM_gaize_, PCM_microsilica_, based on diatomite, gaize and microsilica, respectively.

### 2.2. Methods for Determining the Properties of Raw and Synthesized Materials

The chemical and mineral composition, including the content of the amorphous phase, was characterized using an ARL9900 X-ray spectrometer by X-ray fluorescence (XRF), including the full-profile Rietveld method.

The particle size distribution of powdery silica components of the composite material was carried out using an ANALYSETTE 22 NanoTec laser particle size analyzer (Fritsch, Idar-Oberstein, Germany) by laser backscattering.

The specific surface area of powder materials and the distribution of nanopores were determined using the methods of gas permeability and low-temperature nitrogen porosimetry using a PSH-12 device (Pribory Khodakova, Moscow, Russia) and a Sorbi-M device (Meta, Novosibirsk, Russia) with SoftSorbi-II ver.1.0 software.

The surface adsorption properties of raw materials and synthesized materials, namely the pH of aqueous mineral suspensions, ζ-potential, isoelectric point and Lewis and Brandsted acid–base centers, were investigated by the potentiometric method (pH meter) and electrophoretic mobility method using a Zetatrac laser analyzer (Microtrac MRB, St. Petersburg, Russia) and indicator method using a LEKI SS1207 spectrophotometer (Leki, Moscow, Russia) in the pK_a_ indicator range from −4.4 to +12.8, respectively. Using the method of infrared (IR) spectroscopy using a VERTEX 70 FT-IR spectrometer (Billeric, MA, USA) in the spectral range from 400 cm^−1^ to 4000 cm^−1^, the presence of bonds between inorganic substances in the composition of the composite material was determined.

The study of the microstructural features of powdery raw materials and synthesized materials was carried out using a Mira 3 FesSem scanning electron microscope (Tescan, Brno, Czech Republic) in high vacuum mode (InBeam) using a Schottky cathode of high brightness, with preliminary deposition of chromium on the surface.

### 2.3. Study of the Features of Sols, Gels and Synthesized TiO_2_ Particles

Determination of rheotechnological characteristics of titanium dioxide sols was carried out on a rotational viscometer. The test was carried out after 120 min of stirring the titanium precursor in an alcoholic medium at room temperature. Samples were taken using a cylindrical measuring system.

The method for finding the intrinsic viscosity according to the Huggins Equation (2), namely, the dependence of the intrinsic viscosity on the concentration of the polymer solution, allows one to indirectly estimate the degree of conversion and the molecular weight of titanium oxide structures formed as a result of sol–gel synthesis:(2)$$\eta_{in}=\left[\eta \right]C_{V}+k^{\prime }{\left(\left[\eta \right]C_{V}\right)}^{2}$$
where *η_in_*—inherent viscosity, *η*—polymer solution viscosity, *C_V_*—polymer solution concentration, *k′*—Huggins constant, a constant depending on the interaction of a polymer with the solvent.

The study of the features of titanium dioxide films was carried out using a polarizing microscope. A drop of sol was placed on a glass slide and, after drying at room temperature for 2 min, was examined under transmitted light.

The effect of the type and concentration of surfactants on the properties of colloidal solutions of titanium dioxide was estimated by comparative analysis, as a result of which the average particle diameter of titanium dioxide was determined using the de Brucker or Harden formula (*D_av_*) according to the particle size distribution data obtained on a Microtrac Zetatrac laser particle analyzer.
(3)$$D_{av}=\frac{\sum_{i=1}^{n}w_{i}\cdot D_{i}^{4}}{\sum_{i=1}^{n}w_{i}\cdot D_{i}^{3}}$$
where *i* is the number of the fraction, *D_i_* is the size of the given fraction, *w_i_* is the mass fraction of this fraction.

Measurement of the particle size in the composition of titanium dioxide sols to assess the stability of the sols depending on the HLB surfactant was performed on a Microtrac Zetatrac laser analyzer. For the study, tetrabutoxytitanium was introduced into the prepared surfactant solution in alcohol and mixed for 120 min. The survey was carried out after 30 min of sol preparation.

### 2.4. Determination of Photocatalytic Activity of Materials

To study the photocatalytic activity of the synthesized materials, we adapted the method described in UNI 11259 [22,23], which consists in assessing photocatalytic discoloration—the degradation of an organic dye on the surface of a cement sample with a photocatalyst under the action of ultraviolet radiation.

To determine the photocatalytic activity, samples, tablets with a diameter of 4 cm and height of 5 mm, were prepared with the following compositions: PCM_diatomite_–white cement (WC); PCM_gaize_–WC; PCM_microsilica_–WC. The PCM/WC ratio for all formulations is 1.2/1. The water–solid ratio is 0.8. The samples were examined at the age of 28 days of normal hardening (temperature (20 ± 3) °C and relative air humidity (95 ± 5%). As a control, “photocatalyst-WC” samples were prepared: with industrial photocatalyst AEROXIDE TiO_2_ P25; with TiO_2_ synthesized according to a procedure similar to PCMs (D_av_ from several tens to 300 nm), but without a silica substrate. The dosages of the photocatalysts and the water–solid ratio of the mixture are similar to the samples with PCM.

Then, an aqueous solution of the organic dye rhodamine B (C_28_H_31_ClN_2_O_3_) with a concentration of 4 × 10^−4^ mol/L in an amount of 1 mL was applied to the surface of the prepared samples using a pipette. The samples were kept in a dark place for 30 min (temperature (20 ± 1 °C) at relative air humidity (60 ± 10%), after which they were exposed to ultraviolet irradiation with illumination (2.50 ± 0.25 mW/m^2^). The experimental setup for carrying out photocatalytic studies (Figure 1) excluded light penetration during testing.

The study of the degree of degradation of organic pollutants on the surface of the samples was carried out on the basis of data on the change in color, in particular, the coordinate *a** of the CIELAB color space [24,25], measured using the GNU Image Manipulation Program (GIMP) 2.10.8 software from photographs of the samples.

The determination of the value of the parameter *a** was carried out before placing the samples under ultraviolet radiation, as well as after keeping them under it for 4 h and 26 h.
(4)$$R_{4}\left(\%\right)=100\frac{a_{\left(0h\right)}^{*}-a_{\left(4h\right)}^{*}}{a_{\left(0h\right)}^{*}},\;R_{26}\left(\%\right)=100\frac{a_{\left(0h\right)}^{*}-a_{\left(26h\right)}^{*}}{a_{\left(0h\right)}^{*}}$$
where *R*_4_ and *R*_26_ are the degree of discoloration (degradation) of the organic dye after 4 h and 26 h of exposure to UV radiation, respectively; *a** (0 h), *a** (4 h), *a** (26 h) is the *a** coordinate value of the CIELAB color space, reflecting the color position in the range from green to red, before exposure to UV radiation (0 h) and after exposure to for 4 h and 26 h, respectively.

To assess the photocatalytic activity of the synthesized composites under natural conditions, KBr powder pellets were prepared with an applied and pressed composite material on the surface. After keeping the samples under natural light (with a day length of 8 h) for 5 days, the efficiency of degradation of the pollutant in visible light was investigated using Equation (3).

## 3. Properties of Components and Composition of the Raw Mix for the PCM Sol–Gel Synthesis

### 3.1. Properties of Silica Materials

At the first stage of the study, the chemical and mineral composition of silica raw materials was determined (Table 1) in order to establish the SiO_2_ content, polymorphic modifications and the degree of amorphousness of silica as a criterion for the effectiveness of using this raw material as a substrate for the sol–gel synthesis of titanium dioxide nanoparticles [26].

The investigated silica materials are predominantly composed of an X-ray amorphous phase, ranging from 32.42 to 96.7%, as well as α-quartz and its cryptocrystalline low-temperature modifications: α-cristobalite and α-tridymite. The presence of a diffuse halo in the X-ray diffraction pattern of microsilica indicates the presence of an amorphous phase in the sample, against the background of which low-intensity peaks of the crystalline phase are visible, which could not be identified.

All investigated materials are polydisperse (Figure 2). They contain particles with sizes from 0.15 to 120 µm (for natural raw materials) and up to 400 µm (for technogenic raw materials).

Analysis of the specific surface area of silica raw materials, determined using the gas permeability method, and comparing it with the values obtained using the multipoint BET (Brunauer-Emmett-Teller) method (Table 2), show the difference in readings, the main role in which is played by the presence of pores of nanometer and micrometer sizes. Thus, the natural silica materials diatomite and gaize are characterized by nanometer porosity with pore sizes from 3.5 to 8.4 nm. Microsilica is characterized by porosity with pore sizes from 3.5 to 8.4 nm at 51% and 80 nm at 49% (Figure 3).

Dispersion, in terms of the size of particles and their distribution, cannot fully be a significant criterion of efficiency, since for natural materials it is regulated by the grinding time, and, therefore, it can be changed; for microsilica, which is prone to aggregation, the dispersion values are valid for particle aggregates. The same applies to the values of specific surface area and porosity.

Since the fixation of titanium dioxide depends on the properties of the surface of the silica support, at the next stage, the adsorption surface properties of silica raw materials (pH of an aqueous mineral suspension, ζ-potential, isoelectric point, Lewis and Bronsted acid–base centers) were studied.

The investigated silica materials have a predominantly alkaline reaction of the medium, pH 7.4–9.4. Materials of natural origin (diatomite and gaize) are characterized by an electrokinetic potential in the positive range of +21.1 mV and +14.9 mV, respectively, and microsilica used as a representative of technogenic raw materials has a negative ζ-potential equal to −13.4 mV. For each component, the pH value of the isoelectric point was determined, which characterizes the ability of particles to stick together (Figure 4): for diatomite—8.2; for gaize—9.7; for microsilica—8.0. The most preferable from the point of view of electrokinetic parameters (native ζ-potential, pH and isoelectric point) is the use of a natural silica material—diatomite—as a carrier of the photocatalyst. Further, according to the degree of application, microsilica and gaize are found in the composition of the photocatalytic composite material.

Using the indicator method [27], the quantitative concentration and distribution of acid–base centers on the surface of particles of the studied silica materials were revealed (Figure 5). The largest total number of active centers is noted on the surface of diatomite. The diatomite surface is characterized by the presence of active centers with pKa = −0.29; pKa = +7.15, which correspond to major Lewis centers and Bronsted major centers. The acid–base character of the gaize particle surface is determined by a number of centers with a predominance of peak values at pKa = −0.29; pKa = +2.5; pKa = +7.15; pKa = +12.8. The formation of Lewis main centers can be explained by isomorphic substitution of silicon atoms in the SiO_2_ crystal lattice by aluminum atoms located on the grain surface, as well as by the presence of clay particles of aluminosilicate composition and autogenous films of alkali and alkaline earth metal oxides [28]. According to the Bronsted theory, the predominance of the main centers on the surface of the natural silica carrier occurs due to the shift of the electron density from the atom of the element to the oxygen orbital, which corresponds to the strengthening of the bond in the OH– group [29]. The microsilica component is a material that has donor–acceptor surface properties with pKa = −0.29; pKa = +6.4; pKa = +7.15; pKa = +8.8. The strength of the acid–base centers of the surface of the investigated silica raw material of technogenic origin is much less than that of the samples of natural genetic affiliation.

A scanning electron microscope was used to study the morphostructural features of mineral components and their comparison with the obtained values of physicochemical properties (Figure 6a–f).

Diatomite has a developed surface and has a polydisperse character with particle sizes in the range of 5–50 µm, composed mainly of siliceous valves of diatoms with a developed inner surface, as well as fragments of dispersed material of various shapes and sizes. The silicon skeletons formed as a result of the metabolism of microorganisms retained their porous structure with regular channels ~300–600 nm in diameter.

The microstructure of the gaize is a collection of clay, opal particles and the remains of the skeletons of microorganisms, as well as their conglomerates in the range from a few micrometers to 15 µm. Gaize particles are silicon remnants of microorganisms in the form of shells of diatoms and a certain amount of clay minerals; there are large individuals 10–25 µm in size, the bulk is made up of particles 2–5 µm in size, which are a kind of cementing substance. In a continuous mass, flakes of clay minerals are visible, some of which are X-ray amorphous. The size of individual flakes does not exceed 2–3 µm.

SEM images of technogenic raw materials formed in the process of gas cleaning of technological furnaces in the production of silicon confirm the results of the particle size analysis of the distribution of particles in terms of the polyfraction composition. The entire mass of matter is composed of spherical particles, the size of which ranges from tenths of a nanometer to 1 µm.

Based on the structural features of the studied materials, the results of laser granulometry and porosity are informative only for ranking silica materials with a technologically established rational dispersion in terms of their effectiveness as a carrier in a photocatalytic composite material.

The main results of the physicomechanical and colloidal–chemical properties of various types of silica raw materials with the original dispersion from the manufacturer as a percentage, where the largest indicator of a particular parameter is responsible for 100%, are reflected in the radial diagram (Figure 7).

Taking into account the microstructural features, mineral composition, acidity of the aqueous mineral suspension and the number of adsorption centers by the indicator method, it is possible to predict an increase in the efficiency of silica raw material as a carrier of a nanosized photocatalyst precipitated by the sol–gel method in the following sequence: microsilica → gaize → diatomite.

The final assessment and ranking of the silica raw materials will be carried out based on the results of determining the photocatalytic activity of the synthesized PCM.

### 3.2. Component Composition and Properties of Sols

At this stage of the work, the properties of titanium dioxide sols were studied depending on: the type of alcoholic solvent (ethanol, isopropanol and n-butanol); the concentration of the organo-inorganic precursor; type and concentration of surfactant. For a control in the course of the study, the properties of the system that affect the photocatalytic activity of the final material were selected: the structure of titanium dioxide sols, the morphology of films based on them, the particle size distribution of the synthesized particles.

To obtain uniformly distributed nanosized titanium dioxide particles during the sol–gel synthesis of PCMs, it is necessary to ensure the stability of the sol, which will depend on the following factors: the nature of the solvent, the concentration of the precursor.

Based on the results of the dependence of the specific viscosity of titanium dioxide sol on the concentration of tetrabutoxytitanium in various solvents (ethanol, isopropanol, n-butanol) (Figure 8), it was found that for each alcohol, the limiting concentration of titanium precursor is determined, at which the characteristic viscosity of the reaction mixture reaches its peak value (for ethanol C_V_ = 22% η_sp_ = 45; for isopropanol C_V_ = 18% η_sp_ = 3.5; for n-butanol C_V_ = 23–25% η_sp_ = 0.5) [30].

Based on the studies carried out, it was revealed that in the composition of the reaction mixture, where ethanol is used as an alcohol solvent, with a tetrabutoxytitanium concentration of 22 vol. %, the highest level of polymerization and formation of nanosized titanium dioxide occurs.

Regardless of the type of solvent used (isopropanol, butanol, ethanol), the curing of TiO_2_ sols on a glass substrate leads to the formation of thin films or crosslinked bulk coatings with a pronounced aggregate-like relief (Figure 9).

Cracking of the titanium dioxide film occurs under planar stretching conditions during dehydration of the reaction mixture of the sol. Due to the evaporation of liquid from the surface of the sol, tensile stresses increase, and cracks appear on the rigid surface.

In the medium of a water–alcohol solvent at a low concentration of tetrabutoxytitanium (Figure 9a,e,i), the appearance of spiral structures associated with the process of self-organization is observed, which does not depend on the nature of the solvent and is caused by mechanical compression deformations.

A comparative assessment of the microstructure shows that with an increase in the TBT concentration (Figure 9b–d,f–h,j–l), the solutions in the process of drying pass from the film’s easily shrinkable structures to the volumetric relief aggregate-like coating. When dry, cracking and spontaneous compression of structures occurs, i.e., the predominance of cohesion forces over adhesion forces. It is noted that a low concentration of tetrabutoxytitanium upon evaporation of the solvent makes it possible to obtain thin films of titanium dioxide, and an increase in the concentration in the sol composition leads to an increase in the molecular weight of the oligomer, leading to the structuring and crystallization of titanium dioxide according to the [–Ti–O–]_n_ scheme, which demonstrates a pronounced tendency to aggregation.

Due to the fact that when a titanium precursor is introduced into ethanol at a concentration TBT (22 vol. %), it is possible to obtain the maximum amount of dissolved film oligomer without the precipitation of an aggregate-like precipitate, unlike other alcohols (in isopropanol—18%, in n-butanol—23–25% by volume concentration of tetrabutoxytitanium), so, to obtain composite materials based on the “SiO_2_–TiO_2_” system, it is rational to use 22 vol. % TBT solution in ethanol. Thus, the composition was determined and the optimal ratio of the components of the reaction mixture—TBT:ethanol (1:3)—was revealed.

To prevent the precipitation of a final product uncontrolled in size and shape into an insoluble precipitate, the effect of surfactants of a wide range of hydrophilic–lipophilic balances on the stabilization of titanium dioxide particles in the reaction medium was studied. For this, nonionic surfactants with HLB at a percentage of 15; 4.7; 4.3; 3.7 were studied. To expand the line of HLB at a percentage from 15 to 3.7, mixtures of the corresponding surfactants with HLB equal to 10, 8 and 6 were obtained.

At strong ethanol dilution (>90%) of the Tween-80/TBT mixture, the particle size increases linearly with an increase in the proportion of Tween-80 (Figure 10). However, with a decrease in the concentration of ethanol, Tween-80, being a nonionic water-soluble surfactant, strongly thickens the system and, as a result of which, it stops the growth of particles at a low level. However, when it is diluted, the particle size rapidly increases due to the absence of surfactant barrier shells formed at the titanium dioxide–ethanol interface. This can be explained by the fact that due to the flexible chain oxygenated polyethylene glycol (PEG), which is part of Tween-80, the solubility of surfactants in alcohol is improved, but due to the small hydrophobic fragment of Tween-80, it does not provide steric factors of aggregate stability. The average particle diameter increases linearly with an increase in the Tween-80/TBT ratio at 90% ethanol in the reaction medium, and linearly decreases with a decrease in the Tween-80/TBT ratio at 60% ethanol in the reaction medium. The size and position of the peak (extremum) of the parabola changes: from 60% ethanol at Tween-80/TBT = 0.1 to 85% ethanol at Tween-80/TBT = 1.8. It is possible to achieve a reduction in particle size when using Tween-80 either by reducing the concentration of ready-made solutions (increasing the consumption of alcohol), or by increasing the proportion of surfactants in the system (increasing the consumption of Tween-80).

Obtaining the smallest particle size (Figure 11) can be achieved with the minimum content of the surfactant mixture and a decrease in its HLB level. In some mixtures with Span-60 and Span-80 (with HLB < 8), an almost linear increase in particle size can be observed with increasing surfactant content. In view of the adsorption and binding capacity of the surfactant mixture at high concentrations, the size of the resulting particles increases, and a decrease in the concentration of the surfactant mixture in the reaction medium makes it possible to reduce the average particle size of titanium dioxide. With an increase in HLB above 10, the effect of surfactants on the system decreases for all cases, which indicates the incompatibility of the reaction mixture “TBT–ethanol” with a surfactant with a straight micellar structure (“oil in water”).

The effect of surfactants on the formation of titanium compounds in the reaction medium decreases linearly with decreasing HLB, which leads to a violation of adsorption–solvation equilibrium at the TiO_2_–ethanol interface and decreases the solvation barrier, leading to a sharp increase in the average size of synthesized particles (Figure 12). According to the results obtained [28], it was revealed that the improvement of the aggregate stability of the titanium dioxide particles obtained by the sol–gel synthesis occurs due to a decrease in the molecular weight of the surfactant and upon the transition to saturated fatty acid esters from unsaturated ones (Span-83 → Span-80 → Span-60). Thus, among the reverse micellar surfactants of the Span type (Span-60, Span-80, Span-83) with HLB less than 5 introduced into the reaction mixture, the smallest average size of synthesized TiO_2_ particles (D_av_) can be obtained using surfactants with HLB = 4.7 (Span-60), with a surfactant/TBT ratio = 1–1.1 in the form of the following composition: TBT = 25%, surfactant = 25%, ethanol = 50%.

Thirty minutes after obtaining the sol from the rational composition, the average size of the synthesized titanium dioxide particles (Figure 13), stabilized by a surfactant and their mixture with different HLB levels, was measured.

The polyethylene glycol included in Tween-80 does not provide aggregate stability to sol particles due to the structure of the hydrophobic fragment. It was decided to completely abandon the use of Tween-80, which strongly interacts with the solvent.

Span-60 was chosen as a stabilizer for the titanium dioxide particles obtained in the process of sol–gel synthesis, as the most suitable surfactant, with the use of which the size of the resulting particles is the smallest.

The assessment of the effect of the pH level on the particle size of the reaction mixture was carried out by reducing the initial acid–base level of the titanium dioxide sol (pH = 5.4) by introducing a 2% solution of nitric acid (Figure 14). It was found that even at low surfactant concentrations in the reaction mixture, there is no noticeable change in the average particle size of titanium dioxide relative to the pH level in the range 4.1–5.4.

Due to its nonionic nature, the surfactant is not affected by NO^3−^ ions in the specified pH range. A decrease in pH below 4.1 leads to a loss of stability and stratification of the system due to the saturation of the solution with electrolytes.

Granulometric analysis of titanium dioxide particles (Figure 15), obtained during synthesis from a reaction mixture of a rational composition, showed two peaks at D_av1_ = 43 nm (78%) and D_av2_ = 690 nm (22%).

Using transmission electron microscopy (Figure 16), the average sizes of titanium dioxide particles in the sol composition at the age of 3 days, obtained without and with the use of a stabilizing nonionic substance Span-60, were estimated.

Without adding a surfactant (Figure 16a), the titanium dioxide particles obtained by the sol–gel method reach sizes from 1 to 2 μm and form chains of several spherical formations. The introduction of Span-60 into the reaction mixture makes it possible to obtain titanium dioxide particles with an average size of about 100 nm, combined into aggregates ranging in size from 250 nm to 1 μm (Figure 16b).

Thus, the resulting composition of the reaction mixture (TBT:ethanol = 1:3) was modified to reduce the average size of the resulting particles by partially replacing the solvent with a surfactant. When using surfactants, namely Span-60, in the ratio TBT:surfactant:ethanol = 1:1:2, a sol with a high content of nanosized fractions of titanium dioxide in ethanol is obtained during the synthesis.

## 4. PCM Production Technology by the Sol–Gel Method

The synthesis of titanium dioxide nanoparticles on a silica support during the preparation of a photocatalytic composite material was carried out by the sol–gel method and consisted of four main stages (Figure 17).

Based on the studies of the processes occurring in the “SiO_2_–TiO_2_–sol” system [31], as well as literature data, a phenomenological model of the structure formation of a photocatalytic composite material is proposed (Figure 18).

At the first stage, using a magnetic stirrer at a constant speed of 500 rpm and a temperature of 60 °C for 25 min, Span-60 is homogenized in ethanol. As the temperature rises, the large Span-60 particles break down into smaller ones. The mixing process contributes to the uniform distribution of associations of surfactant molecules throughout the volume of the alcohol medium (Figure 18, Stage I).

The second stage (Figure 18, Stage II) is the introduction of the main components: a titanium-containing precursor and a silica support. At room temperature, titanium dioxide particles are synthesized in the “alcohol–Span-60” reaction medium after the dropwise introduction of the titanium precursor. During stirring for 120 min, the processes of hydrolysis and polycondensation take place, from titanium alcoholate to precipitated particles of titanium oxides. The introduction of a silica component into the reaction mixture promotes the deposition of newly formed TiO_n_ particles on the surface.

During the formation of a micelle, the surfactant particles bind all the water in the system so that the decomposition of the titanium-containing precursor into alcohol and titanium ion occurs on the micelle surface. The dissolution mechanism of tetrabutoxytitanium in a micellar solution means that Ti^4+^ + 4R^−^ particles are displaced to the center of micelles, forming so-called swollen micelles. With an increase in the amount of Ti^4+^ attached to the micellar structure, as well as in the course of hydrolysis and polycondensation reactions, more complex titanium compounds of HO–[TiO]_n_–H are formed. The resulting reaction mixture “alcohol–Span-60–TBT” is characterized by the presence of Ti^4+^ + 4R^−^ formed as a result of dissociation in a water–alcohol solution and subsequent crystallization of H–[TiO]_n_–H into TiO_2_ particles surrounded by surfactant molecules in a dispersion medium alcohol solution. The interfacial tension in micelles presses on the crystallizing titanium dioxide, thereby limiting the size of the titanium formations.

When a silica component is introduced into the system, surfactant molecules are split off from the most stable dispersed TiO_2_ particles, striving to fill the adsorption zone of silica raw materials, thereby capturing and attracting TiO_2_ to the surface layer of the carrier.

At the end of the second stage of the synthesis of titanium dioxide particles and their deposition on the surface of the silica support, the colloidal solution is converted into a suspension of silica particles with the presence on the surface of amorphous titanium dioxide formed from the solidified sol to form a three-dimensional gel network.

The third stage is characterized by the evaporation of the dispersion medium (Figure 18, Stage III) by means of heat treatment in an oven at a temperature of 115 °C, during which the gel aging process takes place. With the removal of the alcoholic component from the dispersion medium, the bonds between neighboring titanium dioxide particles are strengthened due to the coalescence of the sol. Together with the shrinkage of the three-dimensional TiO_n_ network, TiO_2_ aggregates are formed on the surface of the silica component.

As a result of drying the obtained gel, amorphous sol–gel-derived precipitates are formed on the surface of the particles of the silica component of the “SiO_2_–TiO_2_” system.

During the fourth stage, titanium dioxide particles crystallize on the surface of the silica material (Figure 18, Stage IV) and organic components are removed in a muffle furnace at t = 550 °C for 120 min.

During the high-temperature treatment, stresses develop in the titanium dioxide film dried on the surface of the silica support, promoting the formation of cracks and leading to the formation of small particles of synthesized TiO_2_. In addition, firing the dried composite at a high temperature promotes the transition of titanium dioxide from an amorphous to a crystalline structure.

According to the developed technology for obtaining a photocatalytic composite material from the reaction mixture “silica support–titanium dioxide sol”, anatase or rutile crystals are formed on the surface of silica raw material from amorphous TiO_n_ during heat treatment. In order to determine the modification of the titanium dioxide obtained, the PCM was studied only on one of the supports, in particular, diatomite, which showed the best properties as a substrate as a result of ranking. According to the results of X-ray phase analysis of PCM and diatomite, an anatase modification of titanium dioxide was found (9853-ICSD) at 94 wt. % (Figure 19).

Thus, the composite material obtained by the sol–gel synthesis is characterized by the presence of a photocatalytically active anatase on the surface of the silica support.

## 5. PCM Properties

### 5.1. Physicochemical Properties

The formation of –Ti–O–Si– bonds in the “SiO_2_–TiO_2_” system of the synthesized composite materials was estimated using FTIR spectroscopy in the spectral range from 400 cm^−1^ to 1700 cm^−1^ (Figure 20).

According to IR spectroscopy data, all materials contain a peak in the range 1400–1700 cm^−1^, probably caused by water adsorbed on the surface [32]. The broad absorption band in the IR spectrum of TiO_2_ in the region of 500–700 cm^−1^ is attributed to Ti–O vibrations in bound octahedra [TiO_6_] [33]. The spectra of PCMs based on various silica carriers exhibit absorption bands of the –Si– and –O–Si groups characteristic of siliceous structures at 470–474 cm^−1^ (bending vibrations) and 790–800 cm^−1^ (symmetric stretching vibrations) [34]. The spectra contain absorption bands characteristic of titanosilicates: an intense one with a maximum at 1090 cm^−1^ corresponds to asymmetric stretching vibrations, and an average intensity band at 800 cm^−1^ corresponds to symmetric stretching vibrations of [SiO_4_]_∞_ tetrahedra.

Based on the results of IR spectroscopy, it can be concluded that composite materials are characterized by the presence of Si and Ti bonds in various variations with –O and –OH [35,36]. Since the peak characterizing the –Ti–O–Si– bond is not found in the IR spectrum of the PCM, other properties of composite materials should be considered.

Based on the study of the acid–base properties of the surface of photocatalytic composite materials based on various silica carriers (Figure 21), one can judge the activity of the synthesized composite material. The synthesized composites based on various silica components are amphoteric hydroxides and can exhibit both acidic and basic properties.

The deposition of titanium dioxide on the surface of the silica support affects the surface properties of the latter. Comparative analysis showed that the activity of donor–acceptor properties of the PCM_diatomite_ surface relative to the initial composition of diatomite increased at the centers with pKa = +2.5; pKa = +3.46; −pKa = +8.8. The process of precipitation of titanium dioxide from the reaction mixture onto the surface of diatomite leads to the blocking of Lewis acid sites at pKa = −0.29, as well as of the main Bronsted sites with pKa = +7.15; +7.3; +10.5; +12.8.

The surface of PCM_gaize_ is characterized by a high content of Lewis basic and Bronsted acid sites in comparison with the molar in the initial state. After the synthesis of titanium dioxide particles and their attachment to the gaize, an increase in the strength of the active Lewis centers with pKa = −4.4 is observed. There is an increase in activity in the area of Bronsted acid sites with pKa = +1.3; +2.5; +3.46. A number of Bronsted centers of the gaize surface are blocked due to deposition of titanium dioxide on it, which is characterized by a decrease in the strength of activity with pKa = +2.1; +7.15; +8.8; +12.8.

The surface of PCM_microsilica_ is characterized by the presence of acid–base centers of Lewis and Bronsted, and Bronsted acids are most active in centers with pKa = +1.3; +3.46 and Bronsted bases at the centers with pKa = +7.3; +8.8; +12.0. In addition to an increase in the donor–acceptor activity of a photocatalytic composite material based on microsilica, a decrease in the activity of centers is observed: Lewis base at pKa = −0.29, Bronsted base at pKa = +7.15; +10.5, as well as Bronsted acids at pKa = +2.5; pKa = + 6.4.

In terms of the concentration of acid–base centers on the surface of a photocatalytic composite material obtained on the basis of silica raw materials of natural and human-made origin we found the total number of adsorption centers of the natural silica material diatomite (diatomite—140 × 10^−3^ mg·eq/g; PCM_diatomite_—158 × 10^−3^ mg·eq/g). Less active are PCMs based on microsilica and gaize (microsilica—78 × 10^−3^ mg·eq/g; PCM_microsilica_—113 × 10^−3^ mg·eq/g; gaize—80 × 10^−3^ mg·eq/g; PCM_gaize_—107 × 10^−3^ mg·eq/g).

The surface of composite materials (Figure 22) is a relict structure of silica particles covered with spherical particles of titanium dioxide with inclusions of synthesized material in the pore space.

To assess the distribution of titanium dioxide over the surface of the silica support, mapping was performed for Si, Ti, and other elements that make up the PCM.

Based on the results of determining the elemental composition on the surface of photocatalytic composite materials, we found up to 30 wt. % Ti. The deposition of titanium dioxide on diatomite and gaize mainly occurs on the active centers of the surface of silicate particles and in small amounts on particles of aluminosilicate composition, which is part of the silica raw material. After thermal treatment of the reaction mixture “microsilica–titanium dioxide”, aggregates of TiO_2_ particles are formed from a gel with a dense structure, characterized by the presence of individual titanium dioxide particles with sizes up to 250 nm. SEM images of PCM microsilica do not make it possible to distinguish microsilica particles, since their dimensions are in the same range as the formed titanium dioxide particles. Consequently, the gel mass “microsilica–titanium dioxide” (Figure 21c) is a combination of silica particles and newly formed TiO_2_.

### 5.2. Photocatalytic Activity of Composite Materials

The photocatalytic decomposition of the organic dye on the surface of the sample with the industrial photocatalyst AEROXIDE TiO_2_ P25 after 4 h of exposure under ultraviolet radiation was 28% and, after 26 h, 89% (Table 3). This photocatalyst is widely used for the manufacture of self-cleaning products with the photocatalytic decomposition of various types of contaminants. In this regard, the results of the photocatalytic degradation of rhodamine B on the surface of the sample with the photocatalyst AEROXIDE TiO_2_ P25 were taken as a control when compared with the PCM obtained by the described method based on various silica carriers.

After exposure to UV radiation for 4 h, titanium dioxide obtained according to the described technology of sol–gel synthesis (without the introduction of a silica support) provides degradation of the organic dye by 27% and, after 26 h, 91%, which is practically equal to the self-cleaning ability of a sample with an industrial photocatalyst.

The composite material based on diatomite (PCM_diatomite_) after 4 h of ultraviolet irradiation provides degradation of rhodamine B by 25% and, after 26 h, by 86%. Therefore, PCM_diatomite_ folded with anatase fixed on the surface of diatomite is an effective photocatalytic agent.

PCM_gaize_, after 4 h of ultraviolet exposure, ensures the decomposition of the organic dye by 3%. At the end of the experiment, after 26 h of ultraviolet exposure, 57% degradation of the pollutant is observed. It should be noted that, in spite of the low rate of self-cleaning of the sample of the PCM_gaize_ in the time limit of the experiment, in the long term (5 days), the self-cleaning process is completed completely (until the complete removal of rhodamine B), as in the rest of the studied samples.

The photocatalytic composite based on microsilica (PCM_microsilica_), after 4 h of ultraviolet irradiation, provides degradation of the dye by 23% and, after 26 h, by 85%, which is also close to the control sample in terms of the rate of self-cleaning.

After the completion of the irradiation with ultraviolet light, the samples were placed in visible light conditions and, after 5 days, showed almost complete degradation of the organic dye (Table 3). Due to the peculiarities of diatomite and opoka, due to their origin, PCMs obtained by the sol–gel method on their basis are able to sorb dye particles, with their subsequent decomposition by anatase. Therefore, PCM_diatomite_ and PCM_gaize_ are photocatalytic materials with prolonged action.

The most effective, from the point of view of using the photocatalytically active modification of titanium dioxide as a carrier, is diatomite, on the basis of which a composite material is formed with a high content of donor–acceptor centers and a uniform distribution of anatase on the surface of the silica carrier.

## 6. Conclusions

1. It has been shown that by using the sol–gel technology, it is possible to obtain a composite material with nanosized titanium dioxide particles from a reaction mixture consisting of an alcohol solvent, a titanium precursor and a surfactant (2:1:1) deposited on the surface of silica raw materials of various genetic affiliations.

2. Based on the totality of physical and chemical properties, composition, colloidal and chemical indicators (acidity of water–mineral suspension, number of adsorption centers according to the indicator method), taking into account the data of scanning electron microscopy (morphostructural features and qualitative assessment of porosity), an increase in the efficiency of silica raw materials is predicted in the case of using a finished commercial product in the following sequence: microsilica → gaize → diatomite.

3. Based on the results of studies of the polycondensation process in the sol–gel synthesis of titanium dioxide particles, the effect of the nature of the solvent and the concentration of tetrabutoxytitanium in it on the morphology of structures obtained by solidification of titanium dioxide sols was revealed. The concentration limit of TBT (22 vol. %) in ethyl alcohol was determined, before exceeding which there is a strong thickening of the solution associated with the formation of two-dimensional film structures, and then a decrease in viscosity caused by the transition from the film state to bulk three-dimensional particles. The solutions obtained in this way can be used for efficient deposition on the surface of a natural silica support with the formation of new nanoscale growths. The nonionic surfactant Span-60 was chosen as a stabilizer for the titanium dioxide particles obtained in the process of sol–gel synthesis, as the most suitable substance, when the size of the resulting particles is the smallest. The optimal composition of the reaction mixture was modified to reduce the average size of the resulting particles by partially replacing the solvent with a surfactant. When surfactants are used in a given ratio during the synthesis, a solution is obtained with a high content of the nanosized fraction of titanium dioxide in ethanol.

4. For the developed system of TBT:surfactant:ethanol = 1:1:2, it was found that the pH level in the range 4.1–5.4 does not affect the change in the average size of the synthesized titanium dioxide particles in the reaction mixture with the TBT/(surfactant + ethanol) = 1:3.

5. Technological stages have been developed and a phenomenological model of the processes of structure formation of a photocatalytic composite material in the “silica support–photocatalyst” system has been proposed, taking into account the interaction of titanium complexes formed as a result of sol–gel synthesis with the surface of the support of predominantly silica composition by introducing them into the adsorption region of the surface-active substance. The formation of amorphous titanium dioxide particles, the size of which is regulated by a spatially limited nanoreactor formed by a nonionic surfactant, includes the processes of hydrolysis and polycondensation of an organo-inorganic titanium precursor in an alcoholic medium. The initiation of crystallization processes for amorphous sol–gel-derived precipitates promotes the formation of a photocatalytically active anatase deposited on a silica support. The combination of the described processes makes it possible to obtain a composite material that is an active photocatalytic component for self-cleaning materials in various fields of application.

According to the results of the degradation of the water-soluble dye rhodamine B under the action of ultraviolet radiation, it was found that the diatomite-based composite material exhibits the highest activity among the composite materials obtained by deposition of nanosized anatase on a silica support. The indicators of the decomposition activity of the pollutant made it possible to rank the obtained composite materials in terms of the efficiency of their use as components of self-cleaning systems: PCM_gaize_ → PCM_microsilica_ → PCM_diatomite_.

## Figures and Tables

**Figure 1 nanomaterials-11-00866-f001:**
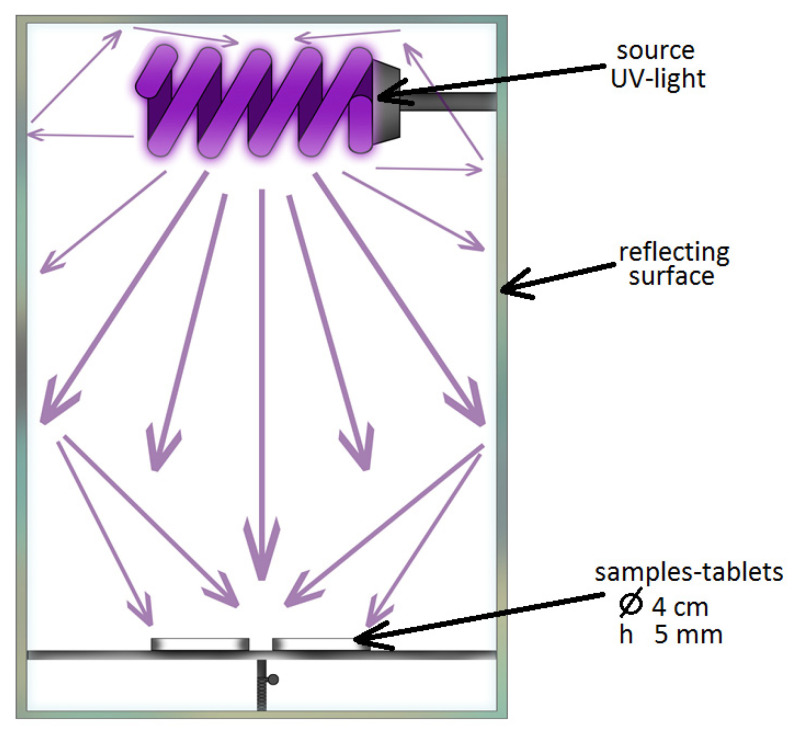
Diagram of the experimental setup for the photocatalytic test.

**Figure 2 nanomaterials-11-00866-f002:**
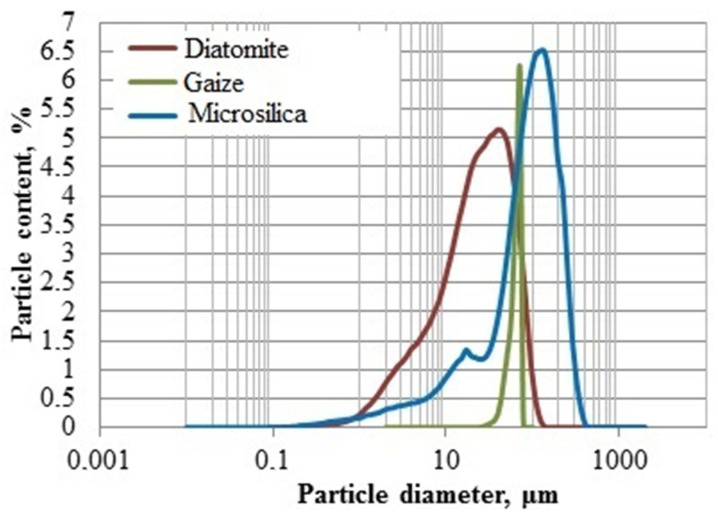
Differential particle size distribution curves of various types of silica materials.

**Figure 3 nanomaterials-11-00866-f003:**
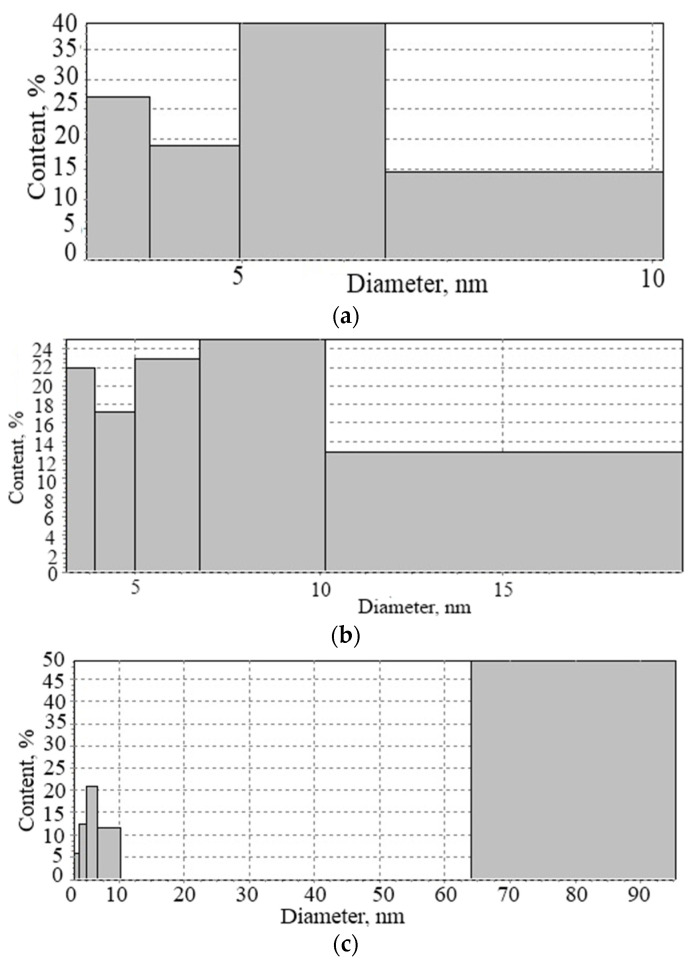
Distribution of nanopores of silica materials: (**a**) diatomite, (**b**) gaize, (**c**) microsilica.

**Figure 4 nanomaterials-11-00866-f004:**
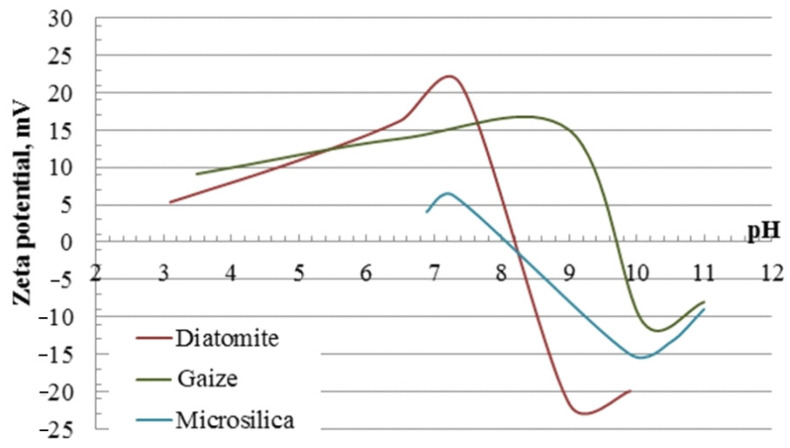
Zeta potential of silica materials of various genesis depending on the pH of the dispersion medium.

**Figure 5 nanomaterials-11-00866-f005:**
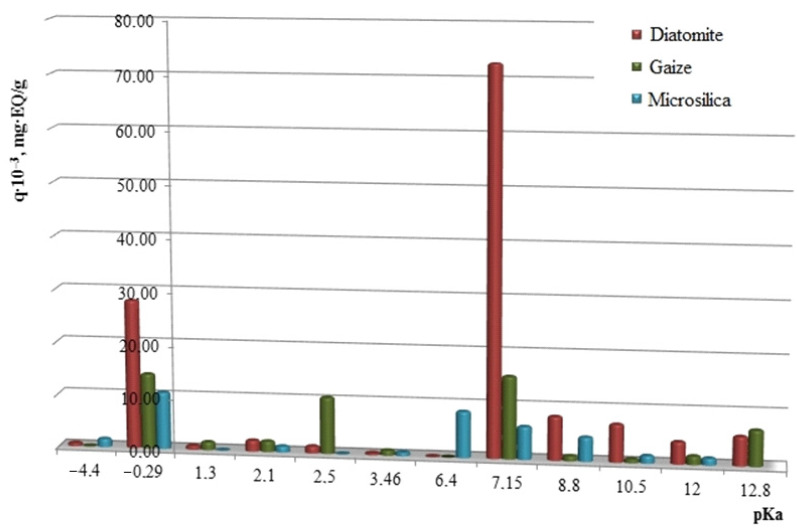
Distribution of adsorption centers on the surface of silica materials.

**Figure 6 nanomaterials-11-00866-f006:**
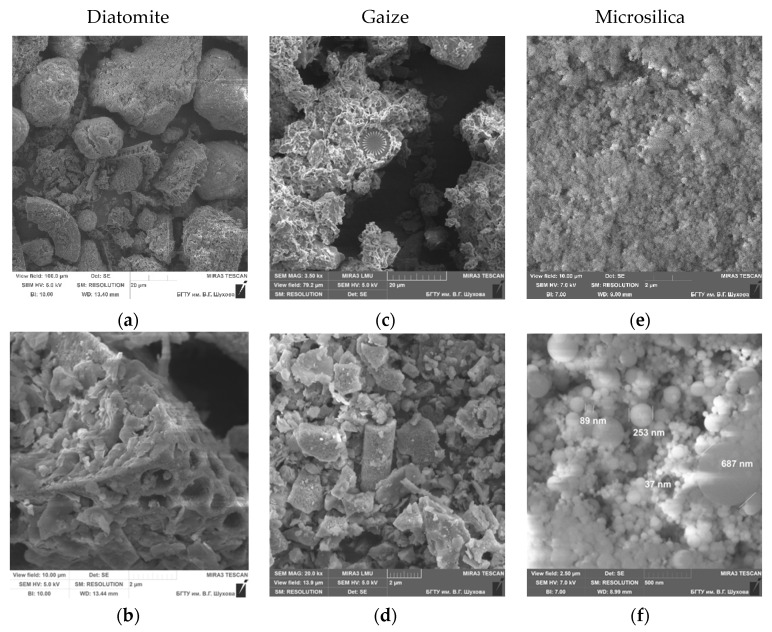
Morphostructural features of the silica materials: diatomite (**a**,**b**), gaize (**c**,**d**), micrisilica (**e**,**f**).

**Figure 7 nanomaterials-11-00866-f007:**
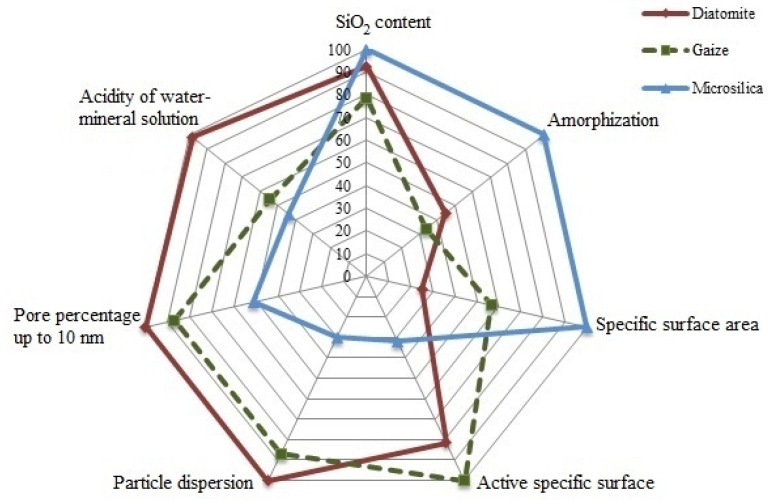
Ranking of the silica materials as photocatalytic composite material (PCM) components according to basic physicochemical and colloidal–chemical properties.

**Figure 8 nanomaterials-11-00866-f008:**
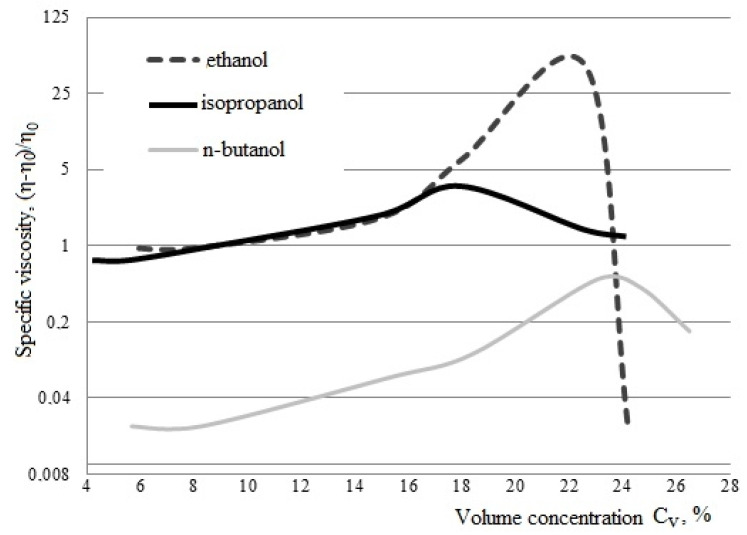
Dependence of the specific viscosity of the titanium dioxide sol on the volumetric concentration of tetrabutyl titanium in various solvents [30].

**Figure 9 nanomaterials-11-00866-f009:**
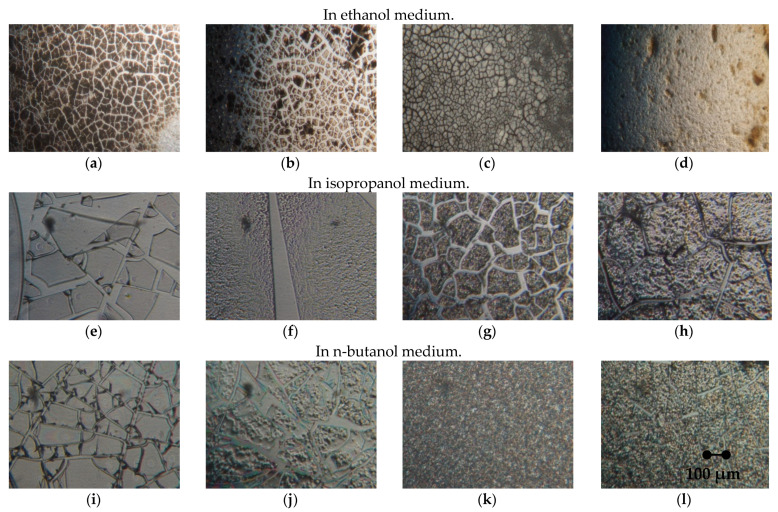
Microstructure of titanium dioxide films formed during drying depending on the volume concentration of tetrabutoxytitanium (TBT) in various solvents: (**a**,**e**,**i**) 8%, (**b**,**f**,**j**) 15%, (**c**,**g**,**k**) 18%, (**d**,**h**,**l**) 22% (at the same magnification).

**Figure 10 nanomaterials-11-00866-f010:**
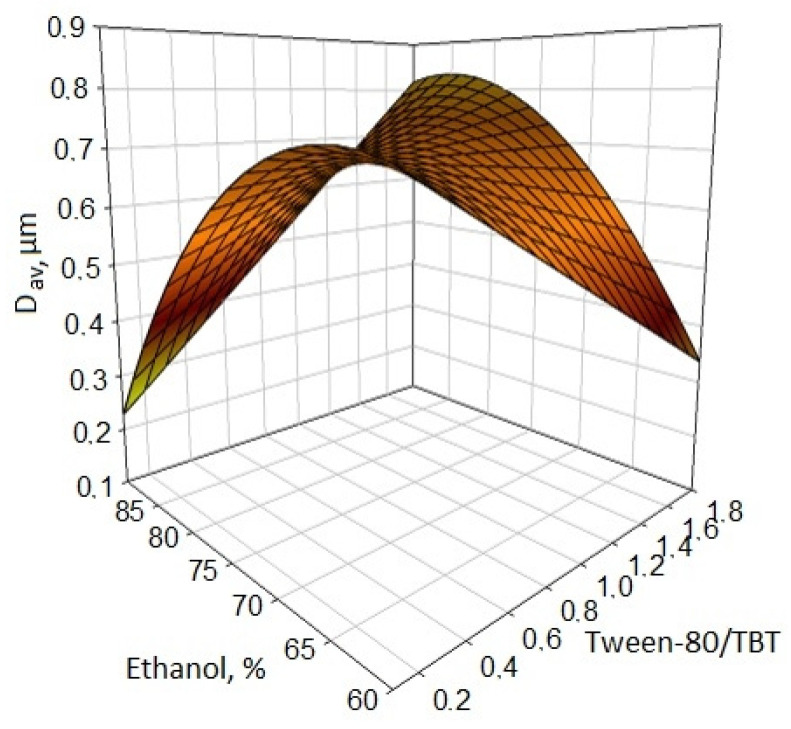
Effect of the Tween-80/TBT ratio in the reaction mixture with various degrees of ethanol dilution on the average size of the titanium dioxide particles obtained.

**Figure 11 nanomaterials-11-00866-f011:**
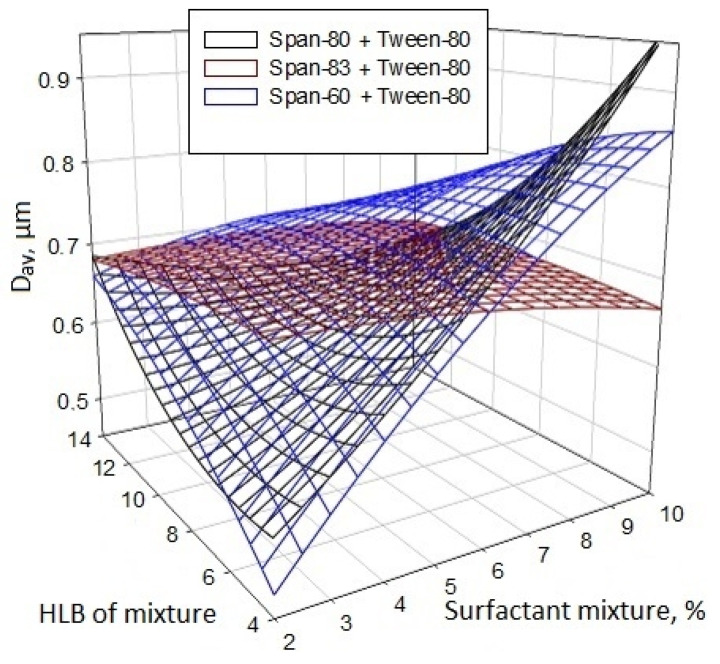
Influence of the composition of the surfactant mixture and its concentration in the TBT/ethanol (1:3) reaction mixture on the average size of the resulting titanium dioxide particles.

**Figure 12 nanomaterials-11-00866-f012:**
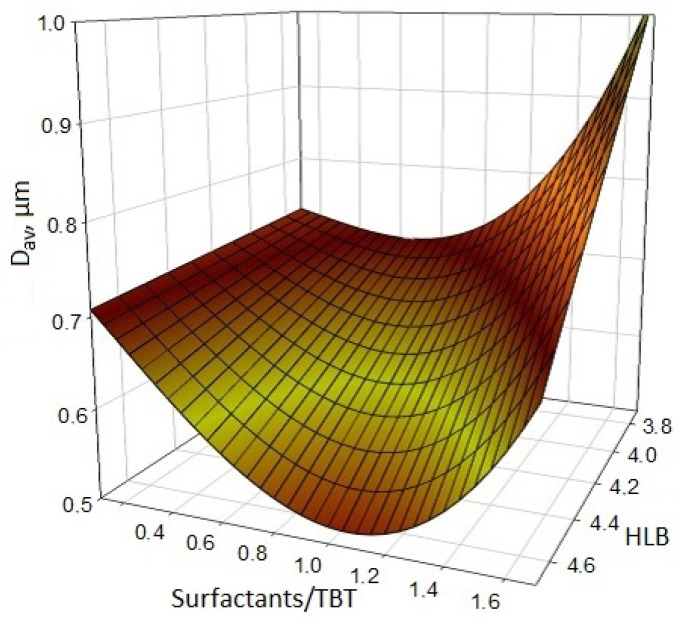
The effect of the type of surfactant (according to the hydrophilic-lipophilic balance (HLB) level) and its mass fraction in the reaction mixture relative to TBT on the average size of the resulting particles [30].

**Figure 13 nanomaterials-11-00866-f013:**
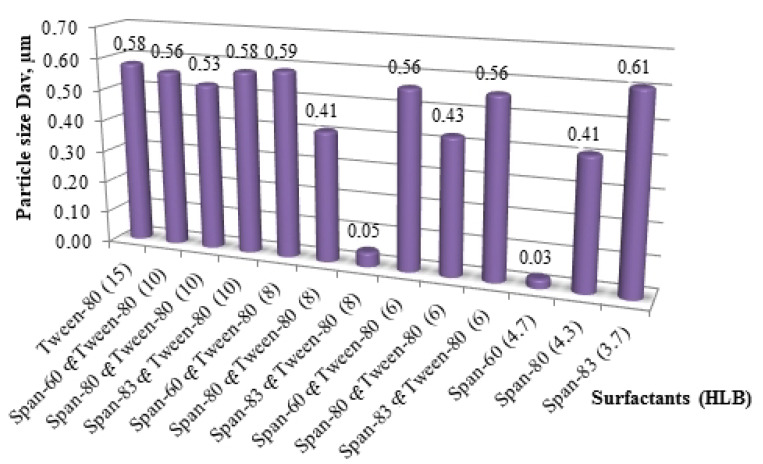
Average values of titanium dioxide particle sizes in the sol composition depending on the type of surfactant and their mixtures.

**Figure 14 nanomaterials-11-00866-f014:**
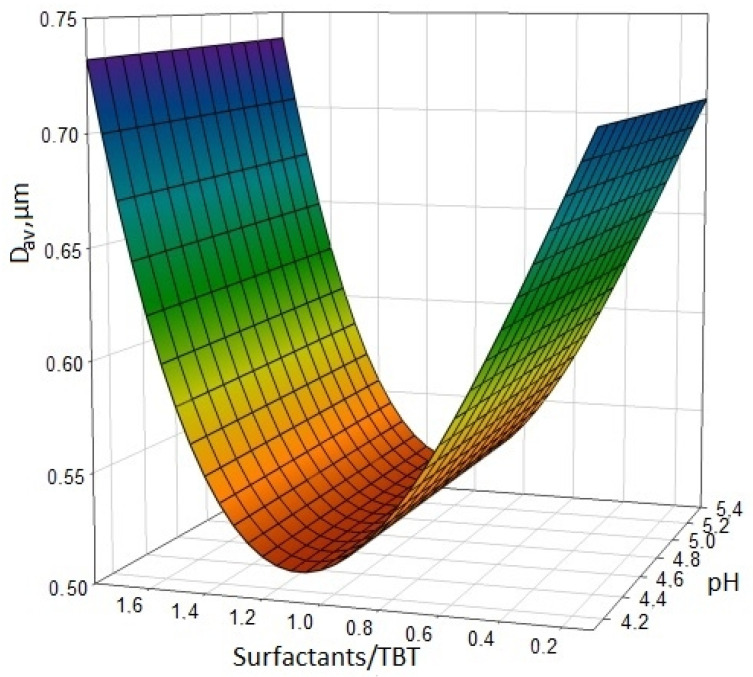
Influence of the surfactant/TBT ratio in the reaction mixture and influence of pH on the average size of the resulting particles of titanium dioxide [30].

**Figure 15 nanomaterials-11-00866-f015:**
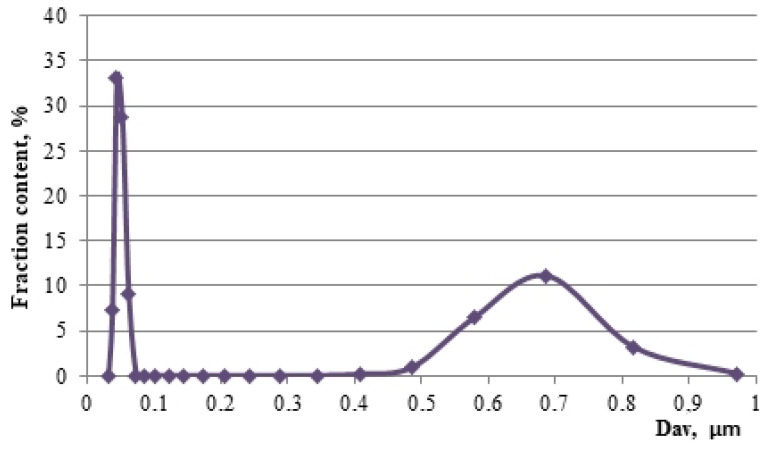
Particle size distribution of titanium dioxide in a reaction mixture of rational composition [30].

**Figure 16 nanomaterials-11-00866-f016:**
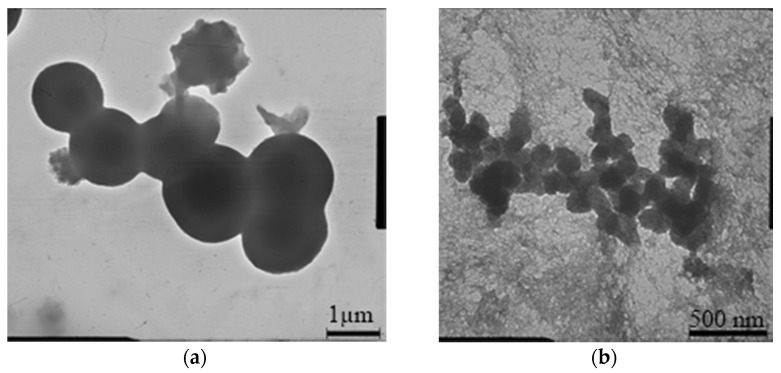
SEM images of the structure of titanium dioxide particles in ash: (**a**) without the addition of surfactants; (**b**) with the addition of surfactants.

**Figure 17 nanomaterials-11-00866-f017:**
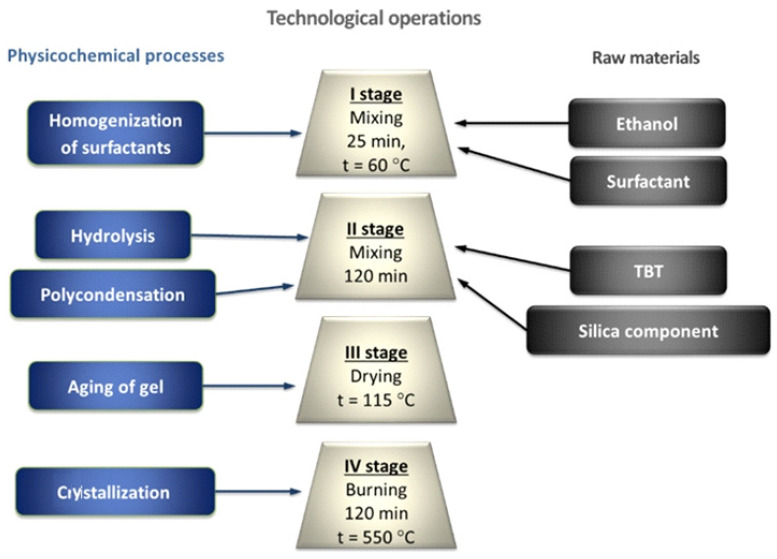
Technological stages and physicochemical processes for obtaining a photocatalytic composite material.

**Figure 18 nanomaterials-11-00866-f018:**
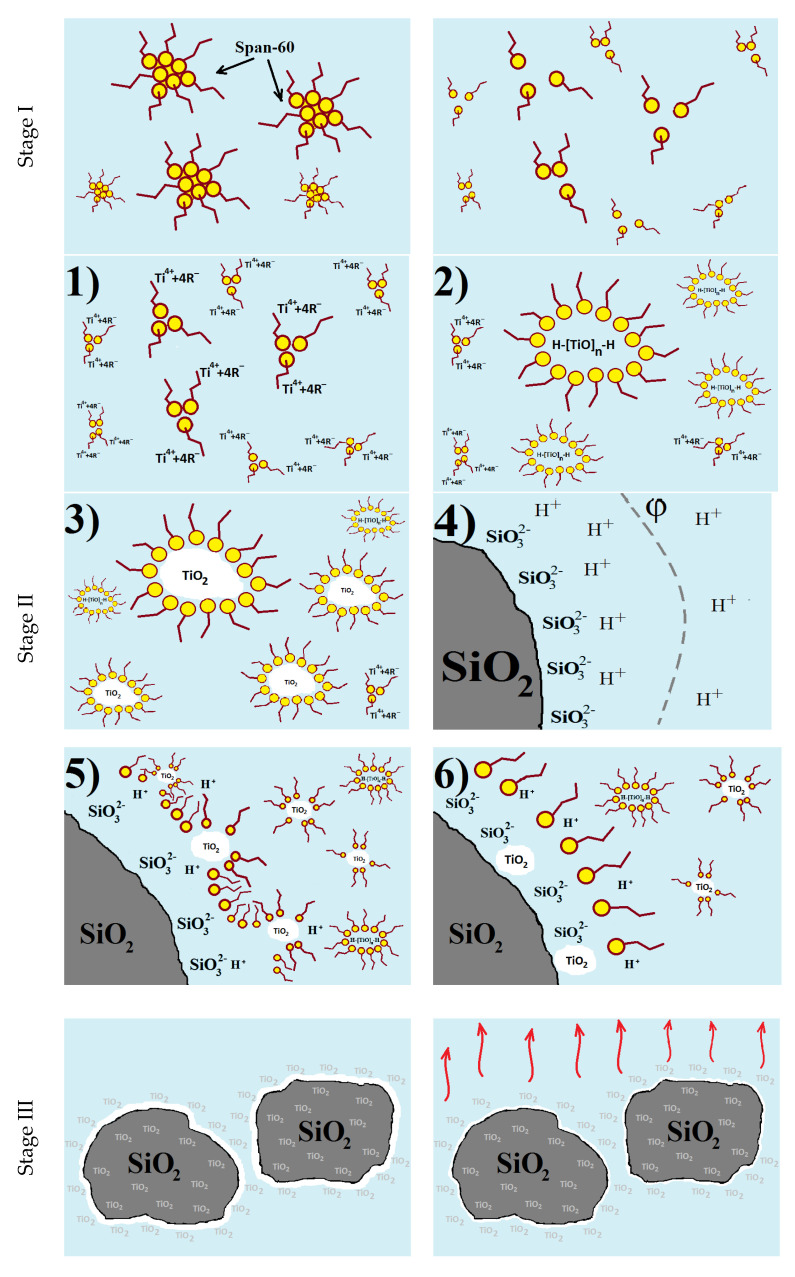
Phenomenological model of structure formation of a photocatalytic composite material.

**Figure 19 nanomaterials-11-00866-f019:**
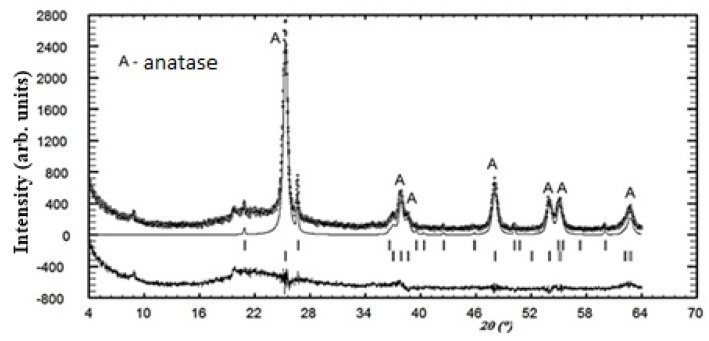
XRD pattern of the PCM_diatomite_.

**Figure 20 nanomaterials-11-00866-f020:**
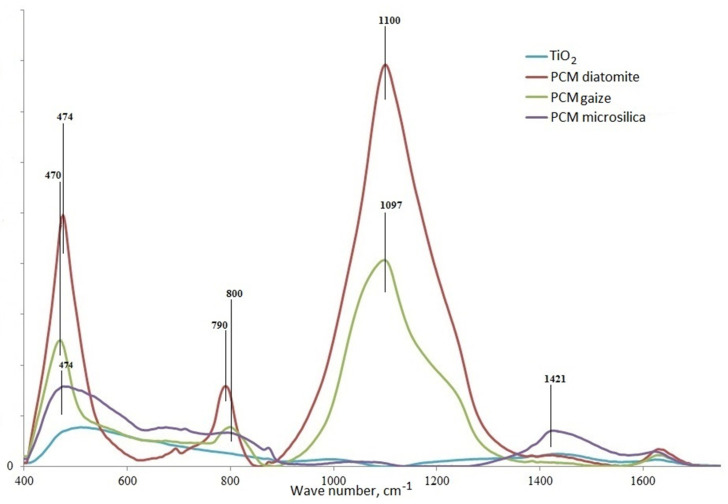
IR spectra of synthesized particles of titanium dioxide and composite materials on silica supports of various types.

**Figure 21 nanomaterials-11-00866-f021:**
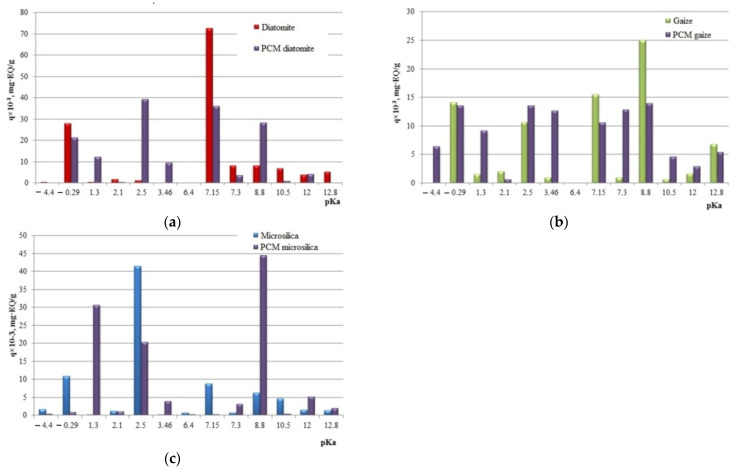
Distribution of adsorption centers on the surface of initial silica materials and PCM synthesized on the basis of: (**a**) diatomite; (**b**) gaize; (**c**) microsilica.

**Figure 22 nanomaterials-11-00866-f022:**
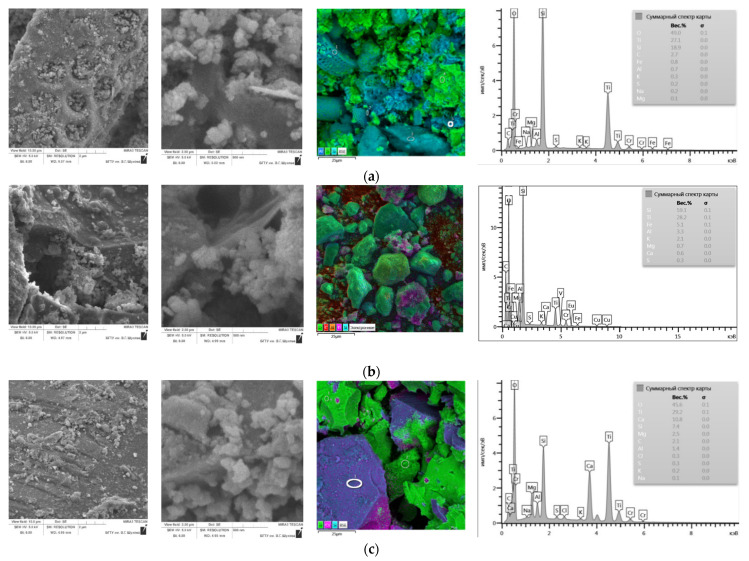
Microstructure and elemental composition of PCM depending on the type of silica support: (**a**) diatomite, (**b**) gaize, (**c**) microsilica.

**Table 1 nanomaterials-11-00866-t001:** The chemical and mineral composition of the starting silica materials.

	Silica Material		
Content of Oxides, wt. %	Diatomite	Gaize	Microsilica
SiO_2_	86.81	73.46	93.09
Al_2_O_3_	5.91	13.26	0.43
Fe_2_O_3_	2.67	4.32	4.00
MgO + CaO	2.76	6.01	0.78
K_2_O + Na_2_O	1.46	1.90	0.54
SO_3_	0.01	0.09	0.07
TiO_2_	0.28	0.67	0.01
P_2_O_5_	0.04	0.06	0.07
Mineral content	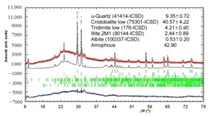	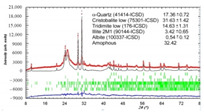	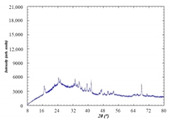
X-ray amorphous phase, %	42.90	32.42	96.7

**Table 2 nanomaterials-11-00866-t002:** Specific surface area of initial silica materials.

Specific Surface Area, m^2^/kg	Initial Silica Materials		
	Diatomite	Gaize	Microsilica
by gas permeability method	1134	2546	4497
by the method of low-temperature nitrogen porosimetry	70,800	86,800	27,700

**Table 3 nanomaterials-11-00866-t003:** Photocatalytic activity of the materials under study.

Test Material	Degradation of Rhodamine B, %		
	UV Exposure Time, Hours		Daylight
	4	26	5 Days
AEROXIDE TiO_2_ P25	28	89	99
TiO_2_	27	91	99
PCM_diatomite_	25	86	94
PCM_gaize_	3	57	93
PCM_microsilica_	23	85	96

## Data Availability

The data presented in this study are available on request from the corresponding author.

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
