# Peer review of "Obtaining and Properties of a Photocatalytic Composite Material of the “SiO2–TiO2” System Based on Various Types of Silica Raw Materials"

_nanomaterials, 2021, doi:10.3390/nano11040866_

Round 1

Reviewer 1 Report

Reviewing of the manuscript: Obtaining and properties of a photocatalytic composite material of the “SiO2-TiO2” system based on various types of silica raw materials.

The manuscript presents the use of matrices with a high content of silica as supports for the anchoring of photocatalytically active TiO2. The objective of this is to achieve that a material with photoactivity properties can be obtained, which, thanks to the low cost and abundance of matrices with silica, can be used as a precursor of construction materials and thus generate self-cleaning surfaces.

A careful analysis of physicochemical properties is presented, which are then correlated with the catalytic performance of the materials, analyzed by their ability to degrade a colorant thanks to the action of UV-vis radiation.

The document is well written and above all it highlights a careful analysis of all the variables analyzed, seeking to establish a connection between the catalytic behavior and the physicochemical properties of the matrices studied. In addition, a large part of the analysis is dedicated to understanding how the photoactive TiO2 phase grows, which is decisive for the application of the materials.

Therefore, I recommend the publication of the manuscript after minor revisions. Some more detailed comments are presented below.

In section 2.2, a more detailed description of the equipment used in every characterization technique should be included, as well as the experimental conditions in every case, instead of a brief description of the properties measured with every technique.

The authors should provide more details about the geometry of the blocks that combine cement and the photocatalytic matrix. In this sense, more information about the experimental setup during the photocatalytic test is required. For instance, the position of the UV-Vis source respect to the surface of the tested material.

A scheme of the experimental setup would be appropriate.

Are the background light of the laboratory composed by residual UV or visible radiations that could influence the measurements? This could be especially determinant if such background light presents fluctuations in the intensity of the radiations. This could be clarified in the scheme of the experimental setup.

Has the possible evaporation of the solvent of the dye any influence during the catalytic test. For instance, alterations in the surface color not by degradation of the dye but for loss of solvent?

According to the IR spectroscopy analysis, no Ti-O-Si interactions were detected so, how does it affect the phenomenological model proposed in figure 17? If no Ti-O-Si connections are observed, there is probably not such a homogeneous distribution of the species as shown in the diagram proposed by the authors.

In the conclusions, the authors stated that: “… Based on the totality of physical and mechanical properties…” However, no mechanical properties were measured. Physico-chemical?

Author Response

Dear Reviewer!

Thank you for your interest in my manuscript. Your valuable comments helped make our manuscript even better. All corrections in the manuscript are highlighted in yellow. Below are the answers to all comments from your first review.

Comment 1:  In section 2.2, a more detailed description of the equipment used in every characterization technique should be included, as well as the experimental conditions in every case, instead of a brief description of the properties measured with every technique.

Response: Clarifications added to the text of section 2.2 (lines 124-125; 127-128; 131-132).

Comment 2:  The authors should provide more details about the geometry of the blocks that combine cement and the photocatalytic matrix. In this sense, more information about the experimental setup during the photocatalytic test is required. For instance, the position of the UV-Vis source respect to the surface of the tested material. A scheme of the experimental setup would be appropriate.

Response: "PCM-WC" samples have a diameter of 4 cm and a height of 5 mm (lines 182-183).

The experimental setup for the photocatalytic test has been added to section 2.4 (Fig. 1).

Comment 3:  Are the background light of the laboratory composed by residual UV or visible radiations that could influence the measurements? This could be especially determinant if such background light presents fluctuations in the intensity of the radiations. This could be clarified in the scheme of the experimental setup.

Response: According to the scheme of the experimental setup (Fig. 1), the interior decoration of the laboratory chamber excludes the penetration of light into it.(lines 195-197)

Comment 4:  Has the possible evaporation of the solvent of the dye any influence during the catalytic test. For instance, alterations in the surface color not by degradation of the dye but for loss of solvent?

Response: Photocatalytic tests are carried out for all samples under the same conditions; therefore, solvent evaporation can be neglected in a comparative analysis of the investigated materials.

Comment 5:  According to the IR spectroscopy analysis, no Ti-O-Si interactions were detected so, how does it affect the phenomenological model proposed in figure 17? If no Ti-O-Si connections are observed, there is probably not such a homogeneous distribution of the species as shown in the diagram proposed by the authors.

Response: The IR spectra of FCMs of various compositions contain peaks characteristic of both silica supports and TiO2. Due to the low titanium dioxide content and the presence of a wide range of vibrations of the Si – O bond (900–1250 cm – 1), the peak of the vibration of the Si – O – Ti bond (900–975 cm – 1) cannot be identified. The Ti-O-Si concentration is below the detection limit by IR spectroscopy.

Comment 6:  In the conclusions, the authors stated that: “… Based on the totality of physical and mechanical properties…” However, no mechanical properties were measured. Physico-chemical?

Response: Agree. Corrected (line 724)

Reviewer 2 Report

This work focuses on the synthesis of a photocatalytic composite material (PCM) by deposition of titanium dioxide particles based on the sol-gel method on a silica support of various types (microsilica, gaize and diatomite).

The properties (chemical and mineral composition, dispersion, specific surface area, porosity, ζ-potential, acid-base properties, and microstructure) of microsilica, gaize and diatomite were studied to assess the effectiveness of using a photocatalytic agent as a carrier. Finally, Rhodamine B was used as a probe molecule in order to check the photocatalytic activity of the developed composite materials.

This is indeed an interesting work. Nevertheless some minor revisions are needed in order to have this manuscript published in MDPI Nanomaterials. 

1.The authors should present more details regarding the synthesis of their samples.

2.The study of the degree of degradation of organic pollutants on the surface of the samples was carried out on the basis of data on the change in its color, in particular, the coordinates a* of the CIELAB color space, measured using the GNU Image Manipulation Program GIMP 2.10.8 software from photographs of the samples. This is not a typical procedure. Usually we use UV-Vis spectroscopy which is an accurate technique. I would expect the authors to verify their approach with UV - Vis. Could they present some references following the methodology using GIMP?

3.Is is quite critical to test the re-usability of the samples, regarding their photocatalytic properties. I suggest the authors to reuse their samples and check their photocatalytic properties at least after 3 cycles of reuse.

This paper could be published in MDPI Nanomaterials after covering the above issues. 

Author Response

Dear Reviewer!

Thank you for your interest in my manuscript. Your valuable comments helped make our manuscript even better. All corrections in the manuscript are highlighted in yellow. Below are the answers to all comments from your first review.

Comment 1: The authors should present more details regarding the synthesis of their samples.

Response: The procedure for the synthesis of a photocatalytic composite material is described in Section 5 (Figures 16, 17); and the technology for preparing samples "PCM-WC" is described in section 2.4 (lines 183-191).

Comment 2: The study of the degree of degradation of organic pollutants on the surface of the samples was carried out on the basis of data on the change in its color, in particular, the coordinates a* of the CIELAB color space, measured using the GNU Image Manipulation Program GIMP 2.10.8 software from photographs of the samples. This is not a typical procedure. Usually we use UV-Vis spectroscopy which is an accurate technique. I would expect the authors to verify their approach with UV - Vis. Could they present some references following the methodology using GIMP?

Response: References  have been added in the text of the manuscript [24, 25] (line 204)

Comment 3: Is is quite critical to test the re-usability of the samples, regarding their photocatalytic properties. I suggest the authors to reuse their samples and check their photocatalytic properties at least after 3 cycles of reuse.

Response:  In this work, the aim was to study the photocatalytic activity of the synthesized materials relative to the commercial photocatalyst AEROXIDE TiO2 P25. The study of the durability of photocatalytic properties will be considered in future works.

Reviewer 3 Report

In this article, a photocatalytic material has been obtained by deposition of titanium dioxide particles synthesized by the sol-gel method on various silica supports. The compounds have been well characterized.
The work is fairly comprehensive and the conclusions are supported by the results. I consider that this work is sufficient to justify the publication in Nanomaterials.

Author Response

Thank you very much for appreciating our manuscript.